# lncRNA 1700101O22Rik and NONMMUG030480.1 Are Not Essential for Spermatogenesis in Mice

**DOI:** 10.3390/ijms23158627

**Published:** 2022-08-03

**Authors:** Yang Zhou, Shijue Dong, Chen Chen, Xiaojun Liu, Xuhui Zeng, Yuan Gao, Xiaoning Zhang

**Affiliations:** 1Institute of Reproductive Medicine, Medical School, Nantong University, Nantong 226019, China; 2013310051@stmail.ntu.edu.cn (Y.Z.); 2113310047@stmail.ntu.edu.cn (S.D.); 2113510009@stmail.ntu.edu.cn (C.C.); 1735718812@stmail.ntu.edu.cn (X.L.); zengxuhui@ntu.edu.cn (X.Z.); 2Experimental Animal Center, Nantong University, Nantong 226001, China

**Keywords:** lncRNA, 1700101O22Rik, NONMMUG030480.1, spermatogenesis, male infertility

## Abstract

Many testis-specific lncRNAs are highly expressed in late spermatogenesis, especially in spermiogenesis. However, their functions and the underlying mechanisms in male fertility are largely unknown. Here, we screened two highly expressed lncRNAs, 1700101O22Rik (O22Rik) and NONMMUG030480.1 (NM480) in testes, to investigate the roles in spermatogenesis using lncRNA knockout (KO) mouse generated by CRISPER/Cas9 technology. Both testis-specific lncRNAs were mainly expressed from secondary spermatocytes to round spermatids, suggesting that they might be involved in spermiogenesis. Phenotypic analysis showed that the deletion of O22Rik or NM480 did not affect the development of testis and epididymis or spermatogenesis. These results were confirmed in both young and middle-aged male mice. In addition, there was no significant difference in sperm morphology and other parameters including concentration and motility between wild type (WT) and KO mice. Fertility tests showed that litter size was significantly lower in O22Rik KO mice compared with WT controls. Although O22Rik did not exert dramatic roles in spermatogenesis, on molecular levels, its surrounding gene expression was disturbed significantly. Gm32773 was decreased; however, Gm32828 was increased in KO mice. In conclusion, lncRNA O22Rik and NM480 are not individually essential for spermatogenesis in mice.

## 1. Introduction

Normal spermatogenesis is crucial for the production of functional sperm and male fertility. It is a complex differentiation process and involves spermatogonium, spermatocyte, and haploid sperm stages [1,2]. Every step of spermatogenesis is strictly and precisely controlled by many genetic and non-genetic factors [3,4]. It is very important to explore new factors that regulate spermatogenesis to reveal the pathological mechanisms of male infertility. A large number of encoding genes regulating spermatogenesis have been well revealed, while little is known about whether non-coding RNAs, especially long non-coding RNAs (lncRNAs), are involved in spermatogenesis in mammals.

LncRNAs, a class of RNAs more than 200 bases long that do not traditionally encode proteins, have been proved to participate in various fundamental biological processes [5]. Generally, they show more tissue-specific expression profiles, less conservation in different species, and relatively lower expression levels compared with mRNAs. Recent studies have shown that lncRNAs expressed in the testis are much more abundant than in other tissues. It is remarkable that the testis-specific lncRNAs are highly expressed in late spermatogenesis, hinting that those lncRNAs may play important roles in male fertility [6,7]. Indeed, lncRNAs were proved to take part in spermatogenesis by deleting hundreds of testicular-specific lncRNAs in the non-mammal drosophila [8]. Kai Otsuka et al. found that lncRNA Start transcripted at Prss/Tessp sites acted on Leydig cells to fine-tune steroid production pathways subtly in the testis in mice [9]. Another testicular germ cell-specific lncRNA, Teshl, preferentially regulated gene expression of the Y chromosome and maintained progeny sex ratio [10].

Mechanically, lncRNAs affect the expression of neighboring or distal genes on transcription and post-transcription levels through different mechanisms [11,12,13,14]. This can orchestrate mRNA-targeted gene expression on the same chromosome in a cis manner or regulate cortical development in trans [15,16]. LncRNAs also directly modulate protein function including enzyme activity. For example, lncRNA ZFAS1 inhibited SERCA2 activity to cause intracellular Ca^2+^ overload [17]. In spermatogenesis, lncRNA Tug1 binds to EZH2 protein to regulate Ccl2 expression, contributing to the blood–testis barrier integrity [18]. Recently, Kastor and Polluks polypeptides encoded by lncRNAs were inserted in the outer mitochondrial membrane and directly interacted with voltage-dependent anion channel to regulate spermatogenesis [19]. Nevertheless, research has shown that single deletion of lncRNAs might have no significant effect on male fertility, suggesting the function of many lncRNAs in spermatogenesis may be redundant or fine-tuning [20,21,22]. Hence, more studies are needed to identify and uncover specific lncRNAs working on spermatogenesis. As it has been reported in the literature, 1700101O22Rik (O22Rik, Gene ID in NCBI: 73575) is expressed exclusively in meiotic prophase and the round spermatid stage in testes [23]. However, it is still unknown how it works in spermatogenesis and male fertility.

In the present study, we chose two lncRNAs highly expressed in the mice testes, O22Rik [23] and NONMMUG030480.1 (NM480, NONCODE Gene: http://www.noncode.org/index.php, last accessed on 17 June 2022), to interrogate whether these lncRNAs play a crucial role in the reproductive process, especially in spermatogenesis based on knockout (KO) mice generated by Cas9 technology.

## 2. Results

### 2.1. O22Rik and NM480 Were Specifically Expressed from Secondary Spermatocytes to Round Spermatids

Firstly, we characterized the two lncRNAs expression profiles in various tissues of mice. The data demonstrated that both lncRNAs were highly expressed in the testis, while in brain, liver, spleen, muscle, intestine, stomach, kidney, lung, and heart, their expression levels were undetectable (Figure 1A) by RT-PCR. Then, specific spermatogenic cells were sorted by flow cytometry to investigate the expression levels of the lncRNAs in the different spermatogenic stages. The O22Rik was the highest expressed in round spermatids; NM480 expression was mainly enriched in secondary spermatocytes and round spermatids (Figure 1B,C). These results were basically consistent with their expression levels in different development times of spermatogenesis (Figure 1D,E).

### 2.2. Construction of O22Rik and NM480 KO Mice

With the purpose of comprehensively and systematically investigating O22Rik and NM480 functions in male reproduction, we generated KO mice using CRISPR/Cas9 technology. As shown in Figure 2A, a pair of gRNAs were designed to cut the genomic DNA loci of O22Rik, which is located on chromosome 12. Then, primers were designed to distinguish different genotypes from WT (550 bp) to KO (720 bp) (Figure 2B) by PCR. Sanger sequencing was further performed for KO and WT bands to confirm the KO position, which was consistent with the expectation (Figure 2C). Finally, the transcript expression level of O22Rik was determined by qPCR in the testis to further verify that the lncRNA was deleted successfully (Figure 2D). With the same idea, NM480 KO mice were also constructed successfully (Appendix A).

### 2.3. The Deletion of O22Rik or NM480 Did Not Affect Spermatogenesis

The testis and epididymis of 12-week-old mice were removed and analyzed. The results showed that there were no differences in the morphology, size, and weight of the testes and epididymides between KO and WT mice (Figure 3A–C). In addition, we smeared sperm on glass slides to observe the morphology of sperm under the microscope. Most of the sperm were normal, and a small number of mouse sperm were headless and tailed, however, the proportion of malformed sperm had no substantial difference compared with WT mice (Figure 3D,E).

Next, we carefully compared the development of spermatogenic cells at each stage between WT and KO mice using PAS staining and found that the spermatogenesis in O22Rik KO mice was not disturbed (Figure 4). Similar results were found in NM480 KO mice (Appendix A). Then, considering previous studies reporting that some KO mice will show a delayed sterility phenotype in middle age and later [24,25], we continued to observe the testicular development at 34 weeks old and found that the spermatogenesis process also appeared normal (Figure 4). This study proved that O22Rik and NM480 were dispensable for spermatogenesis in mice.

### 2.4. The Deficiency of O22Rik or NM480 Did Not Disturb Sperm Motility

Although there was no obvious abnormal in spermatogenesis, we still considered whether lack of lncRNA would have an impact on the sperm motility and function. Therefore, sperm concentration and motility parameters including total motility, progressive motility, average path velocity (VAP), straight-line velocity (VSL), curvilinear velocity (VCL), amplitude of lateral head displacement (ALH), and linearity (LIN) were detected by a computer assisted sperm analysis system (CASA). The data showed that there was no significant difference between KO and WT mice regarding the above parameters both in O22Rik (Figure 5A–H) and NM480 (Appendix A) KO mice. Then, we used a high-speed motion camera to record the movement track of the sperm flagellum [26]. The sperm motility parameters in O22Rik KO mice were the same with WT mice (Figure 5I). Finally, we found that the pregnancy rate of mice was equivalent to WT mice (Figure 5J), but the number of pups per litter in both young and elderly O22Rik KO mice was significantly lower than that in the control groups (Figure 5K).

### 2.5. Deletion of O22Rik Changed Its Surrounding Gene Expressions

Given that lncRNAs could regulate the expression of target genes adjacent on chromosome sites in a cis manner, we further explored whether O22Rik deletion led to dysregulation of Gm32773, Gm32838, and Gm46311 (Figure 6A). The results showed that Gm32773 was significantly down-regulated, whereas Gm32828 was up-regulated in O22Rik KO mice (Figure 6B). These results demonstrated that O22Rik may modulate its neighbor gene expressions at the molecular level.

## 3. Discussion

LncRNA has been a research topic in recent years [27,28]. Although lncRNAs have been considered as a “redundant factor” in the individual regulatory system in previous studies, more evidence has shown that lncRNAs can direct or indirectly participate in cell function regulation and the occurrence of various human diseases [29]. Accumulating studies have characterized many testis-related lncRNAs [30,31,32], but only some pieces of literature have proved that they have substantial impacts on male fertility in vivo [8,10,33]. In fact, most of the lncRNAs’ function is still enigmatic. In this study, we investigate the effects of two lncRNAs, O22Rik and NM480, on spermatogenesis and fertility in mice. Our results showed that these two testis-specific lncRNAs were dispensable for spermatogenesis.

Firstly, we found that O22Rik and NM480 were exclusively expressed in the testis. Further data demonstrated that two lncRNAs were dominantly expressed from secondary spermatocytes to round spermatids, which were validated by the expression of O22Rik and NM480 in the different development stage testes. Their expressions could be detected in the mouse testes at 3 weeks old when spermatogenesis goes into the secondary spermatocytes to round sperm stages. Xiaohui Song et al. used in situ hybridization and revealed that O22Rik was only expressed in sperm cells from step 6 to 9, and the highest level was detected in sperm at step 8 [23]. We proposed that this difference might be attributed to the different experimental techniques presenting different detection sensitivities. The sorted spermatogenic cells by flow cytometry are inevitably contaminated by some traces of other cells, which will lead to the expression of genes in places where they are not normally expressed and could be detectable by qPCR. As our data showed, O22Rik transcript was detected marginally from zygotene to diplotene spermatocytes and elongated spermatids. Overall, the expression levels of NM480 were lower than O22Rik; however, the expression patterns of O22Rik and NM480 are very similar during spermiogenesis, which coincided with the period of transcriptional reactivation during spermatogenesis. Gene transcription in germ cells is reactivated in the late prophase of meiosis and terminates during mid-spermiogenesis [34,35]. This implied that O22Rik and NM480 might exert similar roles in spermiogenesis.

Next, lncRNA KO mice were generated to verify our hypothesis. Unexpectedly, the deletion of these two lncRNAs individually had no overt effects on spermatogenesis, especially spermiogenesis as indicated by the differences in morphology and development of testis and epididymis, sperm morphology, and related motility parameters and patterns between KO and WT mice. Our previous and other studies also found that some lncRNAs (AntiCabs1 [36], Gm31453 [22], Tslrn1 [37], lncRNA5512 [38], and 1700121C10Rik [39]) KO did not reveal specific roles in spermatogenesis and male fertility in mice under standard laboratory conditions. Therefore, some lncRNAs may play redundant, fine-tuning, or environmentally dependent roles and act in response to specific environmental stress factors [37,40]. These findings are also supported by a study on zebrafish [41]. Hence, the investigation of lncRNA function should be mainly focused on the molecular levels, stress conditions, or multiple-gene KO models in the future.

We further evaluated the effects of O22Rik and NM480 on KO male fertility. Although the litter size of O22Rik KO mice was significantly lower than that of the control groups, all KO animals were fertile. This might mean abnormalities in sperm functions such as capacitation or acrosome reaction of O22Rik KO mice; however, this needs to be investigated and confirmed in the future. Previous studies showed that the abnormal reproductive phenotype of some KO mice becomes apparent with age. Many genes in KO mice showed testis premature aging and even progressive subfertility or infertility [42,43,44]. Therefore, 8–34-week-old male mice were used to investigate the effect of age on spermatogenesis and male fertility in O22Rik KO mice. However, no further differences were found in mid-aged mice compared to the young KO mice. Additionally, given that lncRNAs could affect the expression of neighboring genes, Gm32773, Gm32838, and Gm46311 expression levels were detected. Intriguingly, Gm32773 and Gm32828 are also exclusively or enriched expressed in mice testes (Mouse ENCODE transcriptome data), suggesting that they may have similar roles in spermatogenesis. Our data showed that Gm32773 was down-regulated, whereas Gm32828 was up-regulated dramatically, which might make a compensatory effect for the deletion of O22Rik. These results confirmed the viewpoint that O22Rik has mini-trim roles in male reproduction and regulated adjacent gene expressions on the same chromosome in a cis manner. It was reported that O22Rik transcripts were located in the cytoplasm of spermatids rather than the nucleus, which might regulate gene expression at post-transcription levels as competing endogenous RNAs via modulating miRNA-based regulation of gene expression by competing for shared miRNAs [45]. Nevertheless, the specific function and underlying mechanism of these genes affected by O22Rik should be investigated in the future, as they might function together with O22Rik in spermatogenesis and male fertility.

In summary, lncRNA O22Rik and NM480 were exclusively expressed in spermatogenetic cells, especially from the secondary spermatocyte to round spermatid stages. There were no perceptible changes in reproductive phenotypes in NM480 KO mice, demonstrating that it is dispensable for spermatogenesis in mice. For O22Rik KO mice, apart from the litter size of O22Rik KO mice being significantly lower than WT mice, no other significant abnormalities on spermatogenesis and sperm motility and morphology were detected, suggesting O22Rik might also not be essential for spermatogenesis but exert fine-tuning regulation roles in male fertility. This study provides some enlightenment for investigating the roles of lncRNAs in spermatogenesis. Furthermore, it ruled out a gene locus for male infertility and contraceptive studies.

## 4. Materials and Methods

### 4.1. Animals

All the KO mice were produced in the specific pathogen-free (SPF) barrier of the Animal Laboratory Center of Nantong University. C57/BL6 mice were raised under controlled light (12 h light-dark cycle) and temperature (24–28 °C), and were provided with standard food and water. All experiment methods were reviewed and approved by the ETHICS Committee of the Animal Center of Nantong University (SYXK (Su) 2017–0046, 20 July 2017).

Genotyping: Genomic DNA of mouse tail clippings were isolated by dissolving the tissue in 100 µL NaOH (50 mM) at 95 °C for 60 min. Then, the lysate was neutralized with 10 µL Tris-HCl (pH = 6.8). The conventional PCR was conducted to distinguish the genotypes of mice. The products of O22Rik and NM480 WT mice were 550 and 574 bp, respectively, whereas those of the KO mice were 720 and 730 bp, respectively. The primers are listed in Table 1.

Fertility test: 2-month-old sexually-mature male KO mice and WT female mice were caged 1:2 for 3 months to check the fertility and litter size. At least 5 KO and WT genotype male mice were used for the fertility evaluation. The number of pups produced by each female was recorded in detail.

### 4.2. Histology

The testis and epididymis of WT and KO mice were removed intact and fixed in Bouin’s solution. Then they were gently prodded with a syringe needle so that the fixative fluid could penetrate into the testis. After being fixed for 24 h, the tissues were dehydrated with gradient concentrations of ethanol and embedded with paraffin. Hematoxylin-eosin (HE) and periodic acid–Schiff (PAS) were performed according to the standard process as described in a previous study [46]. The stained sections were scanned with Leica Aperio VERSA (Leica, Buffalo Grove, IL, USA). Each experiment was repeated three times in different mice.

### 4.3. RNA Extraction and Quantitate Polymerase Chain Reaction (qPCR)

Total RNA of testes was extracted using MiniBEST Universal RNA Extraction kit (TaKaRa, Tokyo, Japan). Then, RNA was reverse-transcribed to cDNA using a HiScript III RT SuperMix for qPCR (+gDNA wiper) kit (Vazyme, Nanjing, China). qPCR was performed on a LightCycler96 real-time PCR system (LightCycler96, Roche, Mannheim, Germany) using a ChamQ Universal SYBR qPCR Master Mix (Vazyme). The qPCR amplification procedure was as follows: 98 °C for 1 min, 98 °C for 10 s, 60 °C for 20 s, and 72 °C for 20 s, and finally extended for 72 °C for 2 min. The relative abundance of each transcript was calculated by using the 2^−ΔΔC^^T^ method, with target gene expression standardized against that of β-actin. The primers used in this experiment are listed in Table 1.

### 4.4. Analysis of Sperm Morphology, Counts, and Motility

After mice were euthanized, the clean cauda epididymidis was removed and poured into a dish with 3 mL phosphate-buffered solution (PBS, Sigma, St. Louis, MO, USA) preheated to 37 °C. It was cut into pieces and kept warm for 10 min on a shaker. The morphology of sperm was observed and captured under a bright-field microscope after smear. The analyzer settings were as follows: frames acquired: 30; frame rate: 60 Hz; minimum contrast: 30; minimum cell size: 4 pixels; straightness threshold: 50.0%; VAP cutoff: 10 μm/s; VSL cutoff: 0.0 μm/s; magnification: 0.78; temperature: 35.0 °C; and the duration of the tracking time: 0.5 s. A 20 µL sperm sample was placed into a designated chamber with 80 µm depth (Mailang, Nanning, China) to analyze sperm motility parameters and concentration by a computer assisted sperm analysis system (CASA, Hamilton Thorne, Beverly, MA, USA). A minimum of 1000 sperm from at least four different fields were analyzed from each specimen. VAP, VSL, VCL, ALH and LIN of sperm were recorded.

### 4.5. Spermatogenic Cell Sorting

Spermatogenic cells were separated by flow cytometry using Hoechst 33,342 and propyl iodide (PI) staining following the protocol described by Gaysinskaya et al. [47] with minor modifications. Specifically, testicular tissue was incubated for 20 min in Dulbecco’s modified Eagle’s medium (DMEM, Sigma, St. Louis, MO, USA) containing type II collagenase (25 U/mL) and Dispase (0.2 U/mL) at 37 °C in a CO_2_ incubator. Digestion was terminated by adding 10% fetal bovine serum (FBS) and the cells were filtered through 75 μm cell filters. The suspension was washed with DMEM by centrifuging at 900 g for 5 min. Then, 5 µL Hoechst 33,342 (10 µg/mL) was added into the centrifuged samples and incubated in the dark for 20 min. Finally, 2 µL PI (10 mg/mL) and 5 µL DNase I (5 U/µL) were added to the samples for sorting the different stage spermatogenic cells by flow cytometry (FACS Aria3, BD Bioscience, San Jose, CA, USA).

### 4.6. Sperm Motility Trajectory Analysis

The mouse caudal epididymis was removed and bluntly dissociated in 1 mL high-saline solution (HS: 135 mM NaCl, 5 mM KCl, 1 mM MgSO_4_, 2 mM CaCl_2_, 20 mM HEPES, 5 mM glucose, 10 mM lactic acid and 1 mM Na-pyruvate, adjusted to pH 7.4 with NaOH) to release the sperm at 37 °C for 10 min. Then, the sperm suspension was transferred to petri dishes coated with 0.05% polylysine in advance, so that the sperm head was fixed, while the tail could swing freely. The flagellum oscillation of sperm was recorded at 200 fps for 3 s and multiple images were generated using a Hamamatsu digital camera C13440 (Hamamatsu, Tokyo, Japan) equipped in a Nikon microscope. Fuji software was used to synthesize a superimposed image to track the waveform of the sperm flagellum, as previously represented [48].

### 4.7. Statistical Analysis

For all statistical analyses, the results of at least three independent replicates were considered as one case. All data were presented by means ± SEM. The significance of the differences was assessed using a non-paired Student-t test. *p* < 0.05 on both sides was statistically significant, and *p* < 0.01 was highly statistically significant.

## Figures and Tables

**Figure 1 ijms-23-08627-f001:**
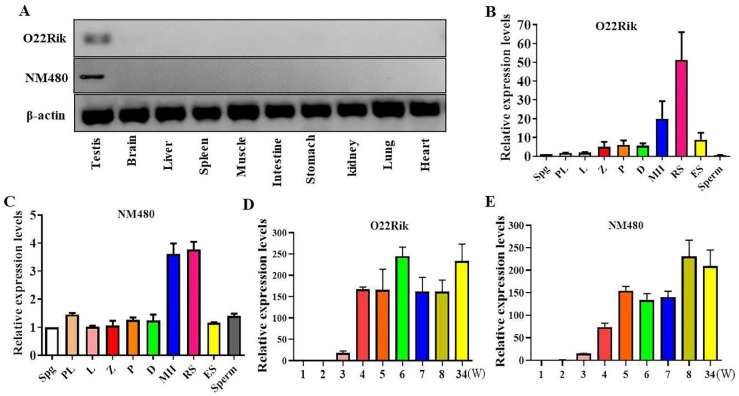
Expression profiles of lncRNA O22Rik and NM480. Total RNA was extracted from each sample using a MiniBEST Universal RNA Extraction kit and transcript expression levels were determined by RT- or qPCR. (**A**) Expression of lncRNA O22Rik and NM480 in various tissues. (**B**) O22Rik and (**C**) NM480 expression levels in the different spermatogenetic cells. Spermatogenic cells were sorted by flow cytometry and their mRNA expression levels were calculated by qPCR assay. (**D**) O22Rik and (**E**) NM480 expression levels in the testis at different stages (1–8, and 34 W (week)) of development. β-actin was used as a loading control to normalize the gene expression levels in different samples. Data were the mean ± SEM (*n* = 3). Spg, spermatogonia; PL, preleptotene spermatocytes; L, leptotene spermatocytes; Z, zygotene spermatocytes; P, pachytene spermatocytes; D, diplotene spermatocytes; MII, meiosis II spermatocytes; RS, round spermatids; ES, elongated spermatids; Sperm, spermatozoa.

**Figure 2 ijms-23-08627-f002:**
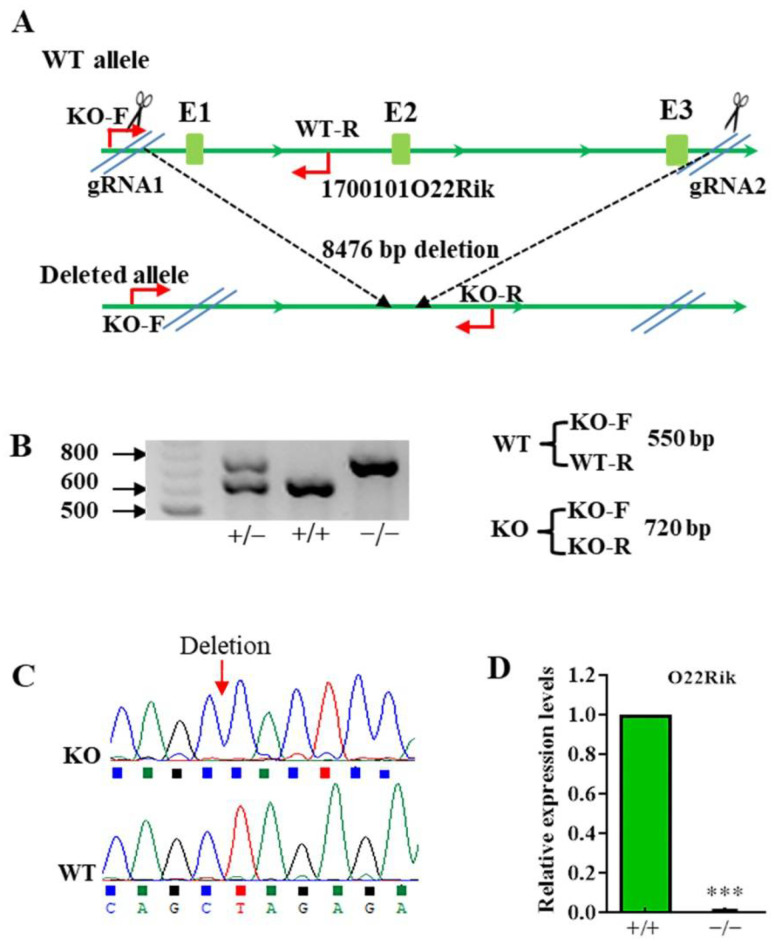
Construction and verification of gene knockout (KO) mice. (**A**) A schematic illustration of the Clustered Regularly Interspaced Short Palindromic Repeat (CRISPR)-Cas9-generated O22Rik KO mice. (**B**) The designed primers were used to identify the genotypes of KO mice. 720bp bands were identified in KO mice and 550 bp bands were detected in WT mice. (**C**) Sanger sequencing was performed using PCR products to verify the successful KO of gene fragment. (**D**) Validation of the O22Rik KO mouse model by qPCR. The O22Rik transcript was lost in KO mouse testes. Total RNA was isolated from the testis and the expression level of target gene was detected by qPCR. β-actin was also used as an internal reference gene to standardize gene expression. Data are the mean ± SEM (*n* = 3). *** *p* < 0.001, compared with the group of WT.

**Figure 3 ijms-23-08627-f003:**
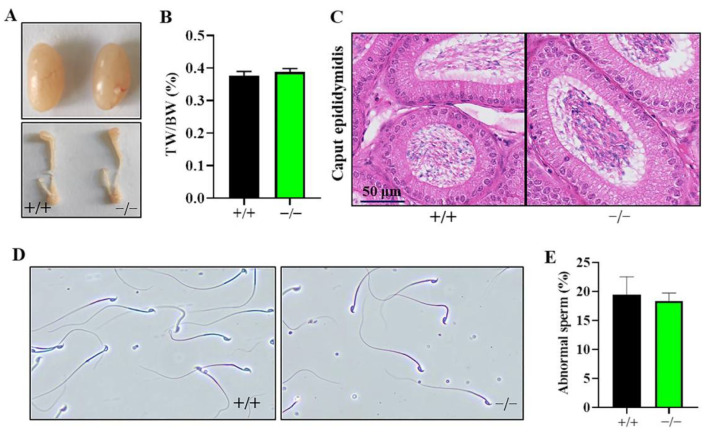
Development and morphology of testis, epididymis, and sperm in knockout (KO) mice. (**A**) The contrast of testis and epididymis between WT and KO mice. (**B**) The ratio of testicular weight to body weight in WT and KO mice. (**C**) Morphological changes of caput epididymis of WT and KO mice. Sections were analyzed by hematoxylin + eosin (HE) staining. (**D**) Sperm morphology analysis in WT and KO groups. Observation of sperm smears were performed under a microscope in 2.5 months old mice. (**E**) The statistical data of abnormal sperm come from (**D**). Data are the mean ± SEM (*n* = 5).

**Figure 4 ijms-23-08627-f004:**
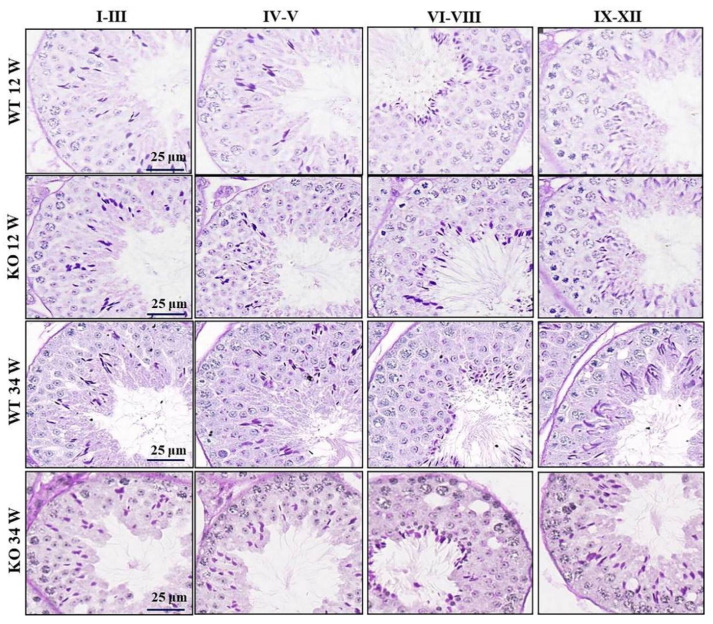
Development and morphology of testes in O22Rik knockout (KO) mice. Histological analysis of testis sections in O22Rik WT and KO mice at 12 and 34 weeks, respectively. Sections were stained with periodic acid–Schiff + hematoxylin (PAS). Different stages show the cycle of the seminiferous epithelium for the mouse.

**Figure 5 ijms-23-08627-f005:**
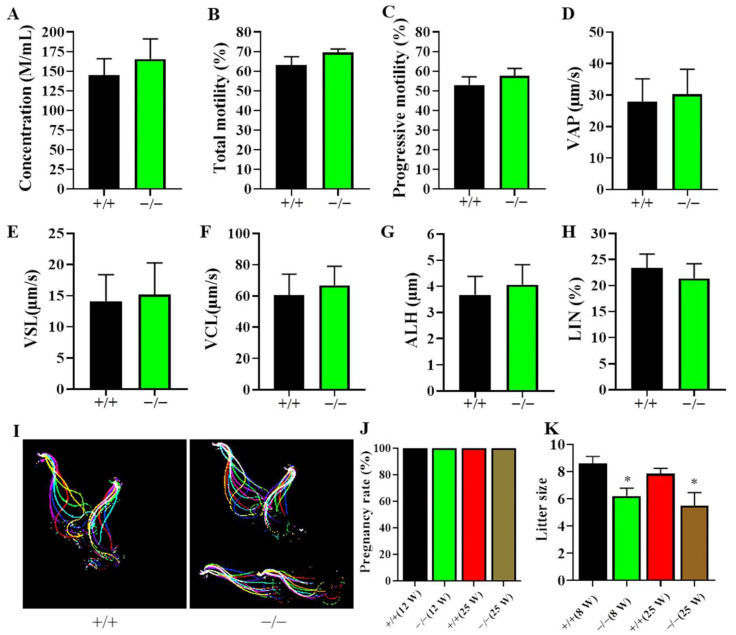
Evaluation of sperm parameters and male fertility in knockout (KO) mice. Sperm were collected from caudal epididymis and measured on a computer assisted sperm analysis system. (**A**) Sperm concentration, (**B**) total motility, (**C**) progressive motility, (**D**) VAP (average path velocity), (**E**) VSL (straight-line velocity), (**F**) VCL (curvilinear velocity), (**G**) ALH (amplitude of lateral head displacement), and (**H**) LIN (linearity) were recorded, respectively. (**I**) The flagellar waveform of WT and O22Rik KO mice. The sperm head was fixed in the dish, and two cycles of sperm tail swing were recorded at a speed of 200 fps, and tracks were superimposed with different colors by FIJI software. (**J**) Pregnancy rates and (**K**) litter sizes were calculated in mating cages (male: female = 1:2) comparing 8- and 25-week-(W)-old male O22Rik KO and WT mice over 3 months. Data are the mean ± SEM (*n* = 5). * *p* < 0.05 compared with WT mice.

**Figure 6 ijms-23-08627-f006:**
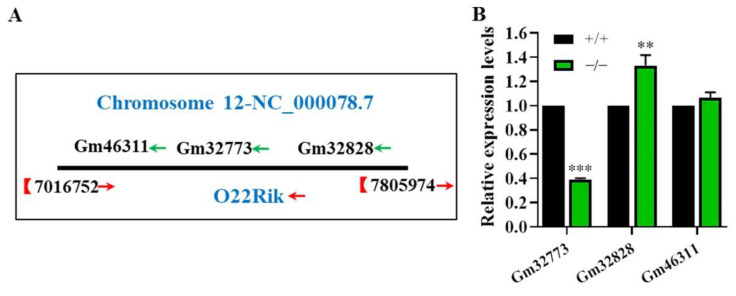
Chromosome location and expression of O22Rik surrounding genes. (**A**) Location diagram of O22Rik adjacent genes Gm46311, Gm32773, and Gm32828 on chromosome 12. (**B**) The expression levels of Gm46311, Gm32773, and Gm32828 in WT and O22Rik. Gene expression was analyzed by qPCR. Data are the mean ± SEM (*n* = 3). ** *p* < 0.01 and *** *p* < 0.001 compared with WT mice.

**Table 1 ijms-23-08627-t001:** Primer information table.

Application	Primer Name	Primer Sequence (5′-3′)
Genotyping	O22Rik-F	TGAATGCTTTGAGAATCCAGGTAGG
	O22Rik-R1	TCCAGTTAGGAAGTGAACAGGAAAA
RT-qPCR	O22Rik-R2	GAACTCATACCACTGCTACGG
	NM480-F1	TCTCCCAGTTGCTTCACTGTCAGG
	NM480-R1	CCCAGGCTTGGGAACTCATTTACA
	NM480-F	TGTATGCACCGAAGGCTGGAGAG
	O22Rik-F	ACCTCTCACCAGAGTGCTTG
	O22Rik-R	GTGTCCTCTCCGTCTGGATG
	NM480-F	TGGCTTGTGTATGCACCGAA
	NM480-R	TAAGCAGACACAGCAGCTCG
	Gm32773-F	CTTGTGACGGTGGTGAGTGA
	Gm32773-R	TTGGCTCACAGTAGGTGGTC
	Gm32828-F	GCTTGTTCTCTCCAACCCTCA
	Gm32828-R	CCAGAAGGCTATGCGTGAGA
	Gm46311-F	CATGCACTGTCCCTGCGAA
	Gm46311-R	ATCATGGACACGAGTTGGGT
	β-actin-F	GGCTGTATTCCCCTCCATCG
	β-actin-R	CCAGTTGGTAACAATGCCATGT

## Data Availability

Data are contained within the Appendix A.

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
