# Peer review of "lncRNA 1700101O22Rik and NONMMUG030480.1 Are Not Essential for Spermatogenesis in Mice"

_ijms, 2022, doi:10.3390/ijms23158627_

Round 1

Reviewer 1 Report

This manuscript clearly demonstrates that two testis-specific LncRNAs O22Rik and NM480 are not essential for mouse spermatogenesis, sperm morphology, sperm motility and fertility. These novel findings are interesting and valuable to the research field of reproductive biology.  

The manuscript is well written. Data analysis and discussion are comprehensive.

This reviewer recommends publishing this manuscript as is.

Author Response

Thank you for your comments and encouragement.

Reviewer 2 Report

The authors Yang Zhou et al. have submitted a manuscript in International Journal of Molecular Sciences (ID: ijms-1801050), entitled “lncRNA 1700101O22Rik and NONMMUG030480.1 are not essential for spermatogenesis in mice)”.

However, the report is a study carried out on two lncRNAs 1700101O22Rik and NONMMUG030480.1; while 1700101O22Rik (022Rik) have about 400 nucleotides, NONMMUG030480.1 (NM480) is not present in the NCBI nucleotides bank, this point must be clarified by the authors.

The authors analyzed their expression in different tissues, interestingly the two transcripts seem to be tissue specific because they are present only in the testis, and particularly in specific spermatogenetic cell. At this aim, they analyzed the lncRNAs identification, expression, validation and targeting mRNA target and produced KO mouse generate by CRISPER/Cas9 technology.

The results showed that the expression of lncRNAs in the testis is not crucial for normal gametogenesis. In addition, the lncRNAs (022Rik) target genes, Gm32773 was decreased whereas Gm32828 increased.

I believe that all data of this study may be useful to increase the knowledge about the lncRNAs; actually, that family of lncRNAs are increasing in number showing new function, however, their role in spermatogenesis remain not completely clear. It is interesting to note that at moment the number of new lncRNAs involved in the testicular activity are increasing.

Finally, I consider the paper suitable for publication in International Journal of Molecular Sciences, after minor revision.

Author Response

However, the report is a study carried out on two lncRNAs 1700101O22Rik and NONMMUG030480.1; while 1700101O22Rik (022Rik) have about 400 nucleotides, NONMMUG030480.1 (NM480) is not present in the NCBI nucleotides bank, this point must be clarified by the authors.

Answer: Thank you for your suggestion. In fact, NONMMUG030480.1 (NM480) is annotated in the lncRNA database NONCODE. (http://www.noncode.org/show_rna.php?id=NONMMUT049240&version=1&utd=0). This information has been provided in our revised manuscript.

Reviewer 3 Report

The authors carried out an interesting study in which they were able to observe that lncRNA 1700101O22Rik and NONMMUG030480.1, could not be essential for spermatogenesis in mica. However,  the present manuscript should be improved.

Summary:

Line 2: slightly lowered ? is this difference significant? if it is so, this should be modified accordingly.

Results:

The figures by themselves should be self-explanatory, therefore this should be improved. If extra info is not added in the figure, it can be added in the figure footnote:
- Indicate the meaning of (W) figures 1D and 1E.
I- ndicate the meaning of 12 W and 25 W in figures 5J and 5K

Line 152-153: Change “motility patern” by “motility paràmetres”.
Line 154-156: Change “slightly less” to “significantly lower”.

Discussion:

Line 223: Change “slightly less” to “significantly lower”. What does this significant difference of lower litter size could mean? I do not understand why the authors downplay this result.

Materials and methods
298-301: Why was the concentration analysis carried out through the CASA system? this system is not the most suitable to evaluate this, since it has a very high error range, etc. In addition, the authors must describe how they performed the motility analysis of the CASA system and the setup used.

Line 304: change, “referring as previously with minor modifications [47]” by “measured following the protocol described by Gaysinskaya et al. [47] with minor modifications”.

Author Response

The authors carried out an interesting study in which they were able to observe that lncRNA 1700101O22Rik and NONMMUG030480.1, could not be essential for spermatogenesis in mica. However, the present manuscript should be improved.

Summary:

Line 2: slightly lowered? Is this difference significant? if it is so, this should be modified accordingly.

Answer: Thank you for your advice. We updated the data of litter size of WT and KO mice and found that their difference was significant. So, we mended this statement in our revised manuscript.

Results:

The figures by themselves should be self-explanatory, therefore this should be improved. If extra info is not added in the figure, it can be added in the figure footnote: Indicate the meaning of (W) figures 1D and 1E.
Indicate the meaning of 12 W and 25 W in figures 5J and 5K
Line 152-153: Change “motility pattern” by “motility parameters”.
Line 154-156: Change “slightly less” to “significantly lower”.

Answer: All above information has been provided in our revision.

Discussion:
Line 223: Change “slightly less” to “significantly lower”. What does this significant difference of lower litter size could mean? I do not understand why the authors downplay this result.

Answer: It’s a good comment. We realize that this is a loose description. Indeed, there was statistical difference in litter size between WT and KO mice in 8-week-old groups. However, in the 25-week-old groups, this difference is not significant at first. During our submission and revision of the article, we continue to supplement and analyze the data of litter size (Increased the sample size to 7) and found that both of groups had significant difference. This might mean abnormalities in some sperm function of KO mice, however, this need to be investigated in future. The related information has been supplemented in the revision.

Materials and methods
298-301: Why was the concentration analysis carried out through the CASA system? This system is not the most suitable to evaluate this, since it has a very high error range, etc. In addition, the authors must describe how they performed the motility analysis of the CASA system and the setup used.

Answer: Thank you for your scrupulous advice and comments. Yes, sperm concentration analysis carried out by CASA has a very high variance. However, our main objective was to analyze the differences between the KO and WT mice. So, each sample was analyzed at least three times and our data collected and calculated from 5-10 mice. In addition, to make sure sperm concentration data is reliable, the previous data were pulled and manually rechecked again. The results are consistent with the findings of the manuscript provided. Of course, in the following studies, we will calculate sperm concentration through the better methods such as the hemocytometer and automatic cell counter after the cell were fixed.

The analyzer settings were as follows: frames acquired: 30; frame rate: 60 Hz; minimum contrast: 30; minimum cell size: 4 pixels; straightness threshold: 50.0%; Average path velocity (VAP) cutoff: 10 μm/s; straight-line velocity (VSL) cutoff: 0.0 μm/s; magnification: 0.78; temperature: 35.0 oC; and the duration of the tracking time: 0.5 s. 20 µL sperm sample was placed into a designated chamber with 80 µm depth (Mailang, Nanning, China) to analyze sperm motility parameters and concentration by computer assisted sperm analysis system (CASA, Hamilton Thorne, Beverly, MA, USA). A minimum of 1000 sperm from at least four different fields were analyzed from each specimen. VAP, VSL, curvilinear velocity (VCL), amplitude of lateral head (ALH) and linearity (LIN) of sperm were recorded.

All these information has been added and updated in our revision. Thanks again.
Line 304: change, “referring as previously with minor modifications [47]” by “measured following the protocol described by Gaysinskaya et al. [47] with minor modifications”.

Answer: This sentence has been revised according to your suggestion. Thank you again.